## [Transparent Peer Review File · Nature Communications]

Single-crystalline $\text{BaSr}_{1-x}\text{TaO}_2\text{N}$ solid-solution photocatalyst with low defect concentrations for solar-driven water splitting

Corresponding Author: Professor Kazunari Domen

Version 0:

Reviewer comments:

Reviewer #1

(Remarks to the Author)

This work presents a single-crystalline $\text{BaSr}_{1-x}\text{TaO}_2\text{N}$ bifunctional photocatalyst for both hydrogen and oxygen evolution reactions. Exponential-tail trap states are formed through post-synthetic high-temperature treatment, facilitating hole participation during the oxygen evolution reaction. The as-prepared BSTON photocatalyst achieved a maximum apparent quantum yield of 13.5% for H_2 evolution and 25.9% for O_2 evolution under 420 nm illumination. However, several important details still require clarification and refinement. Please refer to the following comments.

1. The structure determination of single-crystalline $\text{BaSr}_{1-x}\text{TaO}_2\text{N}$ is insufficient. A more precise structure determination, using high-resolution X-ray diffraction and single-crystal X-ray diffraction, is highly recommended
2. Since the $\text{BaSr}_{1-x}\text{TaO}_2\text{N}$ material contains oxygen elements, it is imperative for the authors to conduct isotope-labeling experiments to confirm the origin of the generated O_2 .
3. The BSTON(TN0.2) exhibits optimal hydrogen evolution activity after post-synthetic annealing at 400 °C and maximal oxygen evolution activity at 900 °C. Considering the use of 400 °C-annealed BSTON(TN0.2) as the hydrogen evolution catalyst and 900 °C-annealed BSTON(TN0.2) as the oxygen evolution catalyst, two critical questions arise: (a) Can such a dual-component system achieve photocatalytic overall water splitting? (b) What solar-to-hydrogen (STH) conversion efficiency could be attained?
4. The photocatalytic stability represents a critical performance parameter that must be rigorously evaluated. Post-reaction structural characterization of single-crystalline $\text{BaSr}_{1-x}\text{TaO}_2\text{N}$ should be systematically conducted.
5. The following recent publications are highly relevant to the current work and should be cited or discussed in the context of your findings: Science 2023, 381, 291; Nat. Energy 2023, 8, 504; Nat. Commun. 2024, 15, 337; Adv. Mater. 2025, <https://doi.org/10.1002/adma.202508693>.

Reviewer #2

(Remarks to the Author)

In the manuscript of "Single-crystalline $\text{BaSr}_{1-x}\text{TaO}_2\text{N}$ solid-solution photocatalyst with low defect concentrations for solar-driven water splitting", Wang et. al. report the preparation of tantalum-based oxynitride photocatalysts using a combination of TaS_2 and Ta_3N_5 as Ta-containing precursors. Owing to the synergistic effect, the as-obtained BSTON shows enhanced crystallinity with reduced structural defects, resulting in the significantly improved activity in photocatalytic hydrogen and oxygen evolution half-reactions. The tantalum-based oxynitride material represents the potential high-performance photocatalyst by considering its narrow band gap and appropriate morphology, and this work provides an important route to prepare high-quality oxynitride crystal. Therefore, I would like to recommend acceptance of this manuscript for publication in Nature Communications after addressing the following issues:

- (1) It is suggested to include the SEM images of TaS_2 precursor and BSTON prepared by using Ta_2O_5 precursor for comparison.
- (2) How about the size distribution of BSTON(TN0.2)?
- (3) Is there any surface vacancy formed on the surface after post-treatment of BSTON(TN0.2) in a reducing hydrogen

atmosphere ?

(4) What happen if co-loading H₂ and O₂ evolution co-catalyst on BSTON(TN0.2) post-treated at 700 oC? Is there any H₂ and/or O₂ evolved in pure water?

(5) Did the authors observed reserve reaction for Cr/Pt or Co loaded BSTON(TN0.2) in H₂/O₂ atmosphere at molar ratio of 2:1?

(6) Can the authors make a brief discussion on the possible reasons that why BSTON can not achieve overall splitting reaction at present?

Reviewer #3

(Remarks to the Author)

I really enjoyed reading this interesting study on oxynitride solid solutions. I have a few minor suggestions for improving this work.

The authors state that the sulphur content is negligible, at least according to XPS, but how much sulphur is actually left in the samples?

The caption of Figure 4 should not be titled 'Photocatalytic water splitting performance of...' as it shows measurements of half reactions with sacrificial agents, no water splitting is performed. Similarly, I think the title should be adjusted as there are no photocatalytic water splitting experiments in this work.

Version 1:

Reviewer comments:

Reviewer #1

(Remarks to the Author)

The author answered the questions well now.

Reviewer #2

(Remarks to the Author)

It is suggested that the revised manuscript be accepted.

Reviewer #3

(Remarks to the Author)

the authors have addressed all my concerns

Response to the reviewers

Reviewer #1 (Remarks to the Author):

This work presents a single-crystalline $\text{Ba}_x\text{Sr}_{1-x}\text{TaO}_2\text{N}$ bifunctional photocatalyst for both hydrogen and oxygen evolution reactions. Exponential-tail trap states are formed through post-synthetic high-temperature treatment, facilitating hole participation during the oxygen evolution reaction. The as-prepared BSTON photocatalyst achieved a maximum apparent quantum yield of 13.5% for H_2 evolution and 25.9% for O_2 evolution under 420 nm illumination. However, several important details still require clarification and refinement. Please refer to the following comments.

1. The structure determination of single-crystalline $\text{Ba}_x\text{Sr}_{1-x}\text{TaO}_2\text{N}$ is insufficient. A more precise structure determination, using high-resolution X-ray diffraction and single-crystal X-ray diffraction, is highly recommended

Response: Thank you very much for your comment regarding the verification of the single-crystalline of the powder sample. However, both single-crystal XRD and high-resolution XRD (HR-XRD) techniques are not suitable for confirming the crystallinity of individual nanoparticles with a diameter of around 50 nm.

These X-ray diffraction techniques rely on coherent diffraction from a well-oriented bulk crystal or epitaxial film, where a single crystallographic orientation dominates within the X-ray illumination area. In contrast, nanosized powders consist of a large number of randomly oriented crystallites. The resulting diffraction pattern therefore represents an ensemble average of all orientations, yielding a typical polycrystalline powder pattern even if each nanoparticle is individually single crystalline. Moreover, because the size of each particle is approximately 50 nm, the incident X-ray beam cannot be confined to a single particle. Therefore, the results obtained in the HR-XRD pattern merely reflect the good crystallinity of the powder ensemble, rather than proving the single-crystalline nature of individual nanoparticles.

To directly verify the single-crystalline nature of individual particles, annular dark-field scanning transmission electron microscopy (ADF-STEM) image of a single BSTON particle was obtained. As shown in **Figure R1**, the individual particle exhibits continuous atomic lattice fringes without any observable grain boundaries or dislocations, confirming that the BSTON photocatalyst particle is a complete single crystal.

Figure R1. Annular dark-field scanning transmission electron microscopy (ADF-STEM) image of an individual BSTON nanoparticle.

2. Since the $\text{Ba}_x\text{Sr}_{1-x}\text{TaO}_2\text{N}$ material contains oxygen elements, it is imperative for the authors to conduct isotope-labeling experiments to confirm the origin of the generated O_2 .

Response: Thank you very much for your comment. Although isotope-labeling experiments could more directly verify whether the evolved O_2 originates from water splitting, unfortunately, our laboratory does not currently have the facilities to perform isotope measurements.

However, for this study, a simpler and indirect approach was used to exclude the possibility that O_2 was generated from the decomposition of the photocatalyst itself. During the photocatalytic O_2 evolution test, the amount of photocatalyst used was 150 mg. If all the oxygen contained in the photocatalyst were released as O_2 , the theoretical amount would be 0.453 mmol. In contrast, the concentration of the Ag^+ sacrificial reagent used was 40 mM, and the maximum O_2 yield under these conditions was about 1.5 mmol — approximately three times higher than the total oxygen content in the photocatalyst.

Moreover, the post-reaction structural characterization (as discussed in comment No. 4) revealed that the structure of the photocatalyst remained unchanged before and after the reaction. Therefore, these results collectively confirm that the evolved O_2 did not originate from the decomposition of the photocatalyst, but rather from the photocatalytic splitting of water.

3. The BSTON(TN0.2) exhibits optimal hydrogen evolution activity after post-synthetic annealing at 400 °C and maximal oxygen evolution activity at 900 °C. Considering the use of 400 °C-annealed BSTON(TN0.2) as the hydrogen evolution catalyst and 900 °C-annealed BSTON(TN0.2) as the oxygen evolution catalyst, two critical questions arise: (a) Can such a dual-component system achieve photocatalytic overall water splitting? (b) What solar-to-hydrogen (STH) conversion efficiency could be attained?

Response: Thank you very much for your comment. Since the BSTON samples exhibited photocatalytic activities for both the hydrogen and oxygen evolution half reactions after post-treatment, we also constructed an overall water-splitting system using samples treated at different temperatures.

Following our previously reported approach, we employed carbon nanotubes (CNTs) as an electron mediator to fabricate a Z-scheme photocatalytic water-splitting system. Unfortunately, no overall water-splitting activity was observed. We believe the main issue arises from the very small particle size of the BSTON photocatalyst, which leads to insufficient interfacial contact with CNTs and partial loss of catalyst particles through the filter paper during filtration process.

We further measured the overall water splitting activity using several redox shuttle systems (e.g., $[\text{Fe}(\text{CN})_6]^{3-}/[\text{Fe}(\text{CN})_6]^{4-}$ and IO_3^-/I^-); however, overall water splitting was unsuccessful. Although the BSTON sample post-treated in H_2 at 400 °C exhibited noticeable hydrogen evolution activity in the presence of redox sacrificial agents, it suffered from severe backward reactions, resulting in poor stability. Meanwhile, the sample reduced at 900 °C in H_2 did not show any oxygen evolution activity in redox-mediated systems. We speculate that this behavior may be attributed to the negatively charged surface of the n-type BSTON, which prevents the adsorption of negatively charged redox species such as $[\text{Fe}(\text{CN})_6]^{3-}$ and IO_3^- , thereby hindering the extraction of photogenerated electrons. Another important issue is the cocatalyst loading. In this work, the cocatalyst loading was optimized based on the half reaction performance, which may not be suitable for the overall water-splitting system. Therefore, further optimization of cocatalyst loading and interfacial design will be necessary in future work to achieve the overall water splitting.

4. The photocatalytic stability represents a critical performance parameter that must be rigorously evaluated. Post-reaction structural characterization of single-crystalline $\text{Ba}_x\text{Sr}_{1-x}\text{TaO}_2\text{N}$ should be systematically conducted.

Response: Thank you very much for your comment. For photocatalytic H_2 evolution, stability was evaluated, and no noticeable performance decay was observed during the initial 40 h of continuous operation (Figure R2). After an extended visible-light photoreaction, the photocatalyst was collected for characterization. XRD patterns confirmed that the perovskite crystal structure of the photocatalyst remained intact before and after the reaction, with no detectable impurity phases (Figure R3), while the XPS spectra of the constituent elements were essentially unchanged (Figure R4), corroborating the stability of both the photocatalyst and the co-catalysts.

For photocatalytic O_2 evolution, Ag^+ was employed as an electron scavenger. The O_2 yield reached saturation within 3 h because Ag^+ was progressively reduced to Ag^0 and deposited on the photocatalyst surface, which precluded long-term stability testing under these measurement conditions. Nevertheless, the $\text{CoO}_x/\text{BSTON}$ sample collected after 3 h showed no change in the crystal structure of the photocatalyst. The XRD patterns confirmed retention of the perovskite phase (Figure R5), except for reflections corresponding to metallic Ag, and XPS revealed the formation of metallic Ag while the chemical states of the other constituent elements remained essentially unchanged (Figure R6). These results further confirm the structural stability of BSTON under photocatalytic operating conditions.

Figure R2. Time courses of photocatalytic H₂ evolution using Cr₂O₃/Pt/BSTON(400 °C).

Figure R3. XRD patterns of the Cr₂O₃/Pt/BSTON (400 °C) photocatalyst before and after a 40-h photoreaction along with the as synthesized BSTON as reference.

Figure R4. XPS Characterization before and after photocatalytic H₂ evolution reaction. The XPS patterns of **a**, Cr 2p, **b**, Pt 4f, **c**, O 1s, **d**, N 1s, **e**, Sr 3d, **f**, Ta 4f acquired from the Cr₂O₃/Pt/BSTON (400 °C) before and after a 40 h photoreaction.

Figure R5. XRD patterns of the $\text{CoO}_x/\text{BSTON}$ ($900\text{ }^\circ\text{C}$) photocatalyst before and after a 3 h photoreaction along with the as synthesized BSTON as reference.

Figure R6. XPS Characterization before and after photocatalytic O₂ evolution reaction. The XPS patterns of **a**, Ag 3d, **b**, Ba 3d, **c**, O 1s, **d**, N 1s, **e**, Sr 3d, **f**, Ta 4f acquired from the CoO_x/BSTON (900 °C) before and after a 3 h photoreaction.

5. The following recent publications are highly relevant to the current work and should be cited or discussed in the context of your findings: Science 2023, 381, 291; Nat. Energy 2023, 8, 504; Nat. Commun. 2024, 15, 337; Adv. Mater. 2025, <https://doi.org/10.1002/adma.202508693>.

Response: Thank you very much for your comment. We have discussed the relevant literature and included the corresponding citations in the revised manuscript.

Reviewer #2 (Remarks to the Author):

In the manuscript of “Single-crystalline $\text{Ba}_x\text{Sr}_{1-x}\text{TaO}_2\text{N}$ solid-solution photocatalyst with low defect concentrations for solar-driven water splitting”, Wang et. al. report the preparation of tantalum-based oxynitride photocatalysts using a combination of TaS_2 and Ta_3N_5 as Ta-containing precursors. Owing to the synergistic effect, the as-obtained BSTON shows enhanced crystallinity with reduced structural defects, resulting in the significantly improved activity in photocatalytic hydrogen and oxygen evolution half-reactions. The tantalum-based oxynitride material represents the potential high-performance photocatalyst by considering its narrow band gap and appropriate morphology, and this work provide an important route to prepare high-quality oxynitride crystal. Therefore, I would like to recommend acceptance of this manuscript for publication in Nature Communications after addressing the following issues:

(1) It is suggested to include the SEM images of TaS₂ precursor and BSTON prepared by using Ta₂O₅ precursor for comparison.

Response: Thank you very much for your comment. The TaS₂ precursor was characterized by SEM and XRD, as shown in the **figure R7**. The commercially purchased TaS₂ exhibits a two-dimensional layered morphology, and its crystal structure is identified as predominantly the 2H-TaS₂ phase.

Figure R7. **a**, SEM image and **b**, XRD pattern for TaS₂ precursor.

When BSTON was prepared using Ta₂O₅ as the Ta precursor, a single-phase BaSrTaO₂N was not obtained; instead, the product was predominantly Ba₃SrTa₂O₉ (**Figure R8**). This is mainly attributed to the higher oxygen content in the Ta₂O₅ precursor, which favors the formation of oxide phases rather than the oxynitride.

Figure R8. **a**, SEM image and **b**, XRD pattern for product synthesized using Ta₂O₅ as the Ta precursor instead of the TaS₂ and Ta₃N₅ mixture.

(2) How about the size distribution of BSTON(TN0.2)?

Response: Thank you very much for your comment. We sincerely appreciate the reviewer's valuable comment. To verify the particle size distribution, dynamic light scattering (DLS) measurements were performed. Although SEM and TEM observations revealed that the size of individual particles was approximately 50 nm, the photocatalyst synthesized by the molten-salt method exhibited inevitable aggregation. As a result, the particle size distribution measured by DLS ranged from 100 to 400 nm, with an average particle size of around 230 nm.

Figure R9. Particle size distribution of the synthesized BSTON.

(3) Is there any surface vacancy formed on the surface after post-treatment of BSTON(TN0.2) in a reducing hydrogen atmosphere?

Response: Thank you very much for your comment. We further characterized the BSTON (H_2 , $900\text{ }^\circ\text{C}$) sample by SEM and did not observe any apparent vacancy on the surface (Figure R10). In addition, BSTON was found to be stable under H_2 reduction at $900\text{ }^\circ\text{C}$. The in-situ TEM observations confirmed that no structural changes occurred during the post-treatment process (Figure S24).

Figure R10. SEM images of the BSTON (H_2 , $900\text{ }^\circ\text{C}$) sample.

(4) What happen if co-loading H₂ and O₂ evolution co-catalyst on BSTON(TN0.2) post-treated at 700 oC? Is there any H₂ and/or O₂ evolved in pure water?

Response: Thank you very much for your comment. When the BSTON was post-treated at 700 °C, both the H₂ and O₂ evolution half reactions were observed. Pt and Co were then loaded onto BSTON, respectively, followed by H₂ reduction at 700 °C. Subsequently, Cr was photodeposited in a methanol aqueous solution, and the photocatalytic overall water-splitting were evaluated in pure water. Unfortunately, no H₂ or O₂ evolution was detected.

(5) Did the authors observed reserve reaction for Cr/Pt or Co loaded BSTON(TN0.2) in H₂/O₂ atmosphere at molar ratio of 2:1?

Response: Thank you very much for your comment. The reserve reaction is a crucial factor in determining whether overall water splitting can be achieved in a one-step excitation process. To examine this, we employed our previously reported method to verify whether different samples exhibit the backward reaction (*Nat Commun* 15, 397 (2024)). A certain amount of H₂ and O₂ gases (in a 2:1 ratio) was injected into the closed circulation system, and the concentrations of H₂ and O₂ were monitored under dark conditions as a function of time. For the CoO_x/BSTON sample, no noticeable changes in H₂ and O₂ concentrations were observed (Figure R11a), even when the initial gas concentrations were increased (Figure R11a), indicating that no backward reaction occurred on this photocatalyst. In contrast, for the Cr/Pt/BSTON sample, both H₂ and O₂ concentrations decreased over time (Figure R11a), and the rate of decrease became more pronounced when higher initial gas concentrations were introduced (Figure R11a). These results clearly demonstrate that a serious backward reaction occurred on Cr/Pt/BSTON, which is one of the main reasons why overall water splitting has not yet been achieved at present.

Figure R11. Reserve reaction under dark conditions. Amounts of gases detected for **a, b** CoO_x/BSTON (H₂, 900°C) and **c, d** Cr₂O₃/Pt/BSTON (H₂, 400°C) with different amount of H₂ and O₂.

(6) Can the authors make a brief discussion on the possible reasons that why BSTON can not achieve overall splitting reaction at present?

Response: Thank you very much for your comment. A discussion has been added in the revised manuscript to address the possible reasons why BSTON has not yet achieved one-step excitation overall water splitting.

BSTON exhibits high activity for both the hydrogen- and oxygen-evolution half reactions after post-treatment, however, this developed material has not yet accomplished overall water splitting. We have made several preliminary attempts, but the exact reasons remain unclear. Based on the current results, we foresee two main directions to realize the overall reaction. First, cocatalyst loading is crucial. In this work, the cocatalysts were optimized for the individual half reactions, which likely promotes a pronounced backward reaction (recombination of H₂ and O₂ to H₂O). Achieving high overall water splitting activity require site-specific cocatalyst deposition at the respective H₂- and O₂-evolution sites, along with further optimization of cocatalyst composition, loading strategy, and interfacial configuration to establish an efficient charge-transfer pathway and suppress the backward reaction. Second, although the density of crystalline defects has been reduced, it remains higher than in our previously reported case of SrTiO₃:Al. Further precursor design, optimization of synthetic conditions, and aliovalent doping will be needed to decrease defect density, mitigate nonradiative recombination, and improve quantum efficiency. These approaches are expected to enable one-step overall water splitting in BSTON. Further investigations are underway to address these issues.

Reviewer #3 (Remarks to the Author):

I really enjoyed reading this interesting study on oxynitride solid solutions. I have a few minor suggestions for improving this work.

The authors state that the sulphur content is negligible, at least according to XPS, but how much sulphur is actually left in the samples?

Response: Thank you very much for your comment. The sulfur composition was examined by XPS. As shown in the **Figure R12**, no S 2p peaks were observed, indicating that the sulfur content was below the detection limit of XPS. Therefore, we concluded that the amount of sulfur in the sample is negligible.

Figure. R12 XPS spectra of Ba 3d, Sr 3d, Ta 4f, O1s, N1s and S 2p for the as-synthesized BSTON(TN0.2) sample.

The caption of Figure 4 should not be titled 'Photocatalytic water splitting performance of...' as it shows measurements of half reactions with sacrificial agents, no water splitting is performed. Similarly, I think the title should be adjusted as there are no photocatalytic water splitting experiments in this work.

Response: Thank you very much for your comment. The original figure caption “Photocatalytic water splitting performance of BSTON samples” has been revised to “Photocatalytic H₂ and O₂ evolution activities of BSTON samples” to describe the results more accurately.